# Synthesis of Gd$_2$Zr$_2$O$_7$ Coatings Using the Novel Reactive PS-PVD Process

Pawel Pędrak, Kamil Dychtoń, Marcin Drajewicz and Marek Góral *

Research and Development Laboratory for Aerospace Materials, Rzeszow University of Technology, Aleja Powstańców Warszawy 12, 35-959 Rzeszów, Poland; pedrak@prz.edu.pl (P.P.); kdychton@prz.edu.pl (K.D.); drajewic@prz.edu.pl (M.D.)
* Correspondence: mgoral@prz.edu.pl; Tel.: +48-17-8653656

**Abstract:** Ceramic topcoats of thermal barrier coatings (TBCs) make it possible to increase the working temperature of the hot sections of jet engines. Yttria-stabilized zirconia oxide (YSZ) is usually used to protect the turbine blades and vanes against high temperature and oxidation. It is necessary to develop new materials which can operate at higher temperatures in a highly oxidizing gas atmosphere. Re$_2$Zr$_2$O$_7$-type pyrochlores are promising YSZ replacements. Usually, they are produced by mixing pure oxides in the calcination process at higher temperatures. In a recent article, the new concept of pyrochlore synthesis during the deposition process was presented. The new technology, called reactive plasma spray physical vapor deposition (reactive PS-PVD), was developed and a Gd$_2$Zr$_2$O$_7$ (GZO) coating was achieved. The reactive PS-PVD process allowed for the use of a mixture of untreated ZrO$_2$ and Gd$_2$O$_3$ powders as reactants, instead of the commercially available gadolinium zirconate powders used in other types of processes. The results of microstructure observations revealed a columnar microstructure in the produced ceramic layer. The phase composition indicated the presence of gadolinium zirconate. Thermal analysis showed a decrease in the thermal conductivity in the range of 700 to 1200 °C of the produced layers, as compared to the layer made of the currently used conventional YSZ.

**Keywords:** thermal barrier coatings; PS-PVD; Gd$_2$Zr$_2$O$_7$; pyrochlore; synthesis; reactive PS-PVD

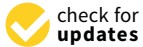



## 1. Introduction

Aluminide coatings and thermal barrier coatings (TBCs) are used widely to protect the hot sections of jet engines and gas turbines against high temperature and oxidation [1–4]. They permit the operating temperatures of the equipment elements to be increased by 80 to 150 °C in comparison to the maximum temperature of these elements without such coatings [1]. The functional properties of thermal barrier coatings depend primarily on the technology used, the chemical and phase composition of their individual layers (e.g., metallic and ceramic topcoats), as well as their thickness [1,5,6]. The ceramic topcoat of thermal barrier coatings is produced mainly by atmospheric plasma spraying (APS) or Electron Beam Physical Vapour Deposition (EB-PVD) processes. The most commonly used ceramic material is yttria-stabilized zirconia oxide (YSZ) [7–9]. Currently, new ceramic materials as alternatives to YSZ are under development [10–13]. A current promising class of materials are pyrochlores with stoichiometric formula Re$_2$Zr$_2$O$_7$, where Re-rare earth elements, mainly from the lanthanides group (from 57 La to 71 Lu). β-zirconium can be used in the ceramic topcoat in thermal barrier coatings [9–12] due to their high phase stability at high temperatures and their thermal properties, such as low thermal conductivity [14]. One of the methods considered as an alternative to electron beam physical vapor deposition (EB-PVD) is plasma spray physical vapor deposition (PS-PVD). This process enables the formation of columnar structures, which is one of its main advantages [15,16]. However, it is possible to control the formation process by changing the parameters, and as a result,

control the coating morphology [17]. Gao et al. [18] described three types of coating structure formation during the PS-PVD process: dense, columnar, and hybrid. Mauer et al. [19] analysed the influence of the composition of plasma gasses and the powder feed rate on the structure of YSZ coatings formed by the PS-PVD method. The new modification of the PS-PVD process is called plasma spray chemical vapour deposition, where liquid precursors for coating synthesis were developed by Gindrat et al. [20]. Recently Qiao et al. [21] proposed the synthesis of $Bi_2O_3$ film using the PS-CVD process. Different novel types of TBC materials produced by the PS-PVD method were developed. Zhang et al. [22] proposed the modification of seven YSZ TBCs by adding Al to the coating, which resulted in the formation of alumina oxide particles. Zhao et al. [23] developed a $La_2Ce_2O_7$ coating using the PS-PVD process. The microstructure [24], deposition mechanism [25], and formation process of this type of coating was deeply investigated. Recently, environmental barrier coatings (EBCs) based on ytterbium monosilicate were developed using the PS-PVD process [26]. The properties of this type of coating in different conditions, such as water vapor, were investigated [27]. Gadolinium zirconate is only one type of pyrochlore used in the production of ceramic layers in the PS-PVD process [28]. The different types of powder preparation methods and their influence on columnar structures were investigated. A comparative study of the properties of double-layer TBCs containing $Gd_2Zr_2O_7$ (GZO) and YSZ with a single-layer pyrochlore coating was also conducted [29]. In our previous research, we investigated the process parameters on the structures of YSZ coatings in the PS-PVD process [30]. The different types of bond coats for ceramic layers produced by the PS-PVD process were also developed [31–33]. Based on Refs. [23–29], the feedstock materials for new types of ceramic coatings were synthetized before deposition. In this paper, we present our new concept of reactive PS-PVD technology with the synthesis of materials from base oxides during the deposition process.

## 2. Materials and Methods

The Inconel 713C alloy was used as a base material. The aluminide bond coat was produced using the chemical vapor deposition method based on parameters described in [34]. The aim of the research was to synthesize gadolinium zirconate using reactive PS-PVD examine the influence of process parameters on the structure and thermal conductivity of the obtained coating. Pure zirconia ($ZrO_2$) and gadolinia ($Gd_2O_3$) oxides were used as feedstock materials for powder preparation. They were mixed in proportions (ratio: $Gd_2O_3$ ~60 wt.%, $ZrO_2$ ~40 wt.%) according to the properties for the synthesis of gadolinium zirconate (GZO). Powders were mixed with PVA alcohol and spray dried. The deposition process was conducted using the LPPS-Hybrid system (Oerlikon Metco, Pfäffikon, Switzerland) in the Research and Development Laboratory for Aerospace Materials at Rzeszow University of Technology. A typical O3CP plasma torch was used in the experiment. The prepared powder mixture (comprising both zirconia and gadolinia oxides) was fed using two 60CD powder feeders. The experimental parameters were selected based on our previous research [30,33] and Refs. [28,29]. The conventional YSZ columnar layer was also deposited using the parameters presented in [33] for comparison. The experimental parameters are presented in Table 1.

**Table 1.** Parameters used for the production of ceramic layers of YSZ and gadolinium zirconate in the PS-PVD experimental processes.

| Process Name | Power Current (A) | Argon Flow (NLPM) | Helium Flow (NLPM) | Powder Feed Rate (g/min) |
|---|---|---|---|---|
| YSZ-2200A | 2200 | 35 | 60 | 2 |
| GZO-1800A | 1800 | 35 | 60 | 2 |
| GZO-2000A | 2000 | 35 | 60 | 2 |
| GZO-2200A | 2200 | 35 | 60 | 2 |

The microstructure of the obtained samples and their thickness were examined using a Hitachi S-3400N SEM microscope (Tokyo, Japan). The elemental mapping was conducted using a Phenom XL SEM microscope (Ambler, PA, USA) equipped with an EDS detector. Analyses of the phase composition of a reference sample and the produced ceramic layers were performed using an ARL X'TRA X-ray diffractometer (Thermo Scientific Corporation, Waltham, MA, USA) (Cu K$\alpha$ radiation Bragg–Brentano geometry, value of the angle 20°–90°). For identification of the phase components, the ICDD-PDF4-2019 crystallographic database was used.

The specific heat of the gadolinium zirconate powder was determined using the Netzsch (Selb, Germany) STA 449 F3 Jupiter cpDSC device using differential calorimetry in accordance with the ASTM E 1269 and DIN 51 007 standards. The obtained values of the specific heat were analysed depending on the temperature, ranging from 700 to 1100 °C. The heating speed was 10 °C/min. The powder for the specific heat analysis was obtained by firing graphite with GZO layers produced in the process of GZO-2200A at 1000 °C for 12 h and then ground in a ball mill.

Analysis of the thermal diffusivity was performed using a Netzsch LFA 427 (Selb, Germany), in the temperature range from 700 to 1100 °C, and a 50 mL/min flow of argon. Before the measurement of thermal diffusivity, the samples were covered with a graphite layer. A two-layer model and the Cape–Lehman method were used for analysis. The substrate with a metallic interlayer was adopted as the first layer, while the second layer was the created ceramic layer. The thermal diffusivity of the substrate was then determined from samples used in the production process of the ceramic layers.

## 3. Results

### 3.1. Microstructure and Chemical Composition

The first stage of research on the produced layers consisted of a microstructure analysis. The obtained microstructure for the layer made of $Gd_2Zr_2O_7$ (GZO) (Figure 1a–c) was very similar to that of the layer made from yttria-stabilized zirconium oxide (YSZ) (Figure 1d), a conventionally used material in PS-PVD processes [16,30]. The ceramic layers had thicknesses of, respectively, 118 +/− 27 μm (GZO-1800A), 125 +/− 7 μm (GZO-2000A) and 127 +/− 5 μm (GZO-2200A). The columns grew perpendicularly to the surface of the substrate, like the ceramic strips made by the gadolinium zirconate coating achieved in the conventional PS-PVD process by Li et al. [28], and were very similar in shape to the columns obtained in the EB-PVD process [35]. The columns in the deposited ceramic layer were characterized by the presence of additional branches and a large number of open and closed pores between the growing columns and the branches [36]. Additionally, with the increase of the plasma power current, 1800→2000→2200 A, the disappearance of the fine fraction from the material was observed (Figure 1a–c), similar to results obtained in [37]. In the structure of the coating, small spheroidal particles of GZO were visible. They are probably formed as a result of secondary crystallization of material observed in longer spray distance of PS-PVD processes. A decreasing number of spheroidal particles with decreasing plasma power current was observed (Figure 1a–c). The elemental mapping of Zr, Gd, and O indicates the uniform distribution of these elements on the cross section of the coating (Figure 2).

### 3.2. Phase Composition

In the phase composition of the obtained coatings (Figure 3), we determined that there were three phase components: $Gd_2Zr_2O_7$ zirconate (marked as CGZO in the chart, ICDD card no. 04-005-5736), $ZrO_2$ oxide (marked as MZO, ICDD card no. 01-070-8739), and $Gd_2O_3$ oxide (marked as CGO in the chart, ICDD card no. 01-073-6442). It was observed that with an increase of the plasma gun's current intensity, the intensity of reflections from $Gd_2O_3$ (e.g., (222)) and $ZrO_2$ (e.g., (111)) oxide decreased. This suggests that there was a decrease in the number of unmelted particles in the starting powder mixture due to changes in the current intensity, which was not observed in the coatings that were produced using calcinated $Gd_2Zr_2O_7$ powder, as reported by Liu [28].

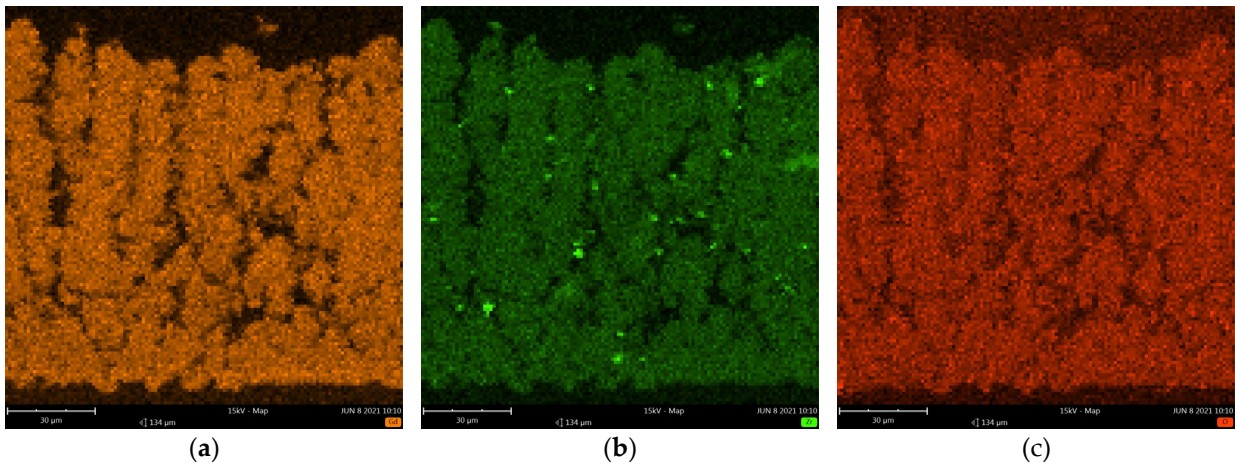

**Figure 1.** Microstructures of $Gd_2Zr_2O_7$ (GZO) layers made with different power current values: (**a**) GZO-1800A, (**b**) GZO-2000A, (**c**) GZO-2200A and microstructure of conventional YSZ-2200A layer (**d**).

**Figure 2.** Elemental mapping of Gd (**a**), Zr (**b**) and O (**c**) distribution in GZO-2200 A coating synthetized in Reactive PS-PVD process using 2200 A power current.

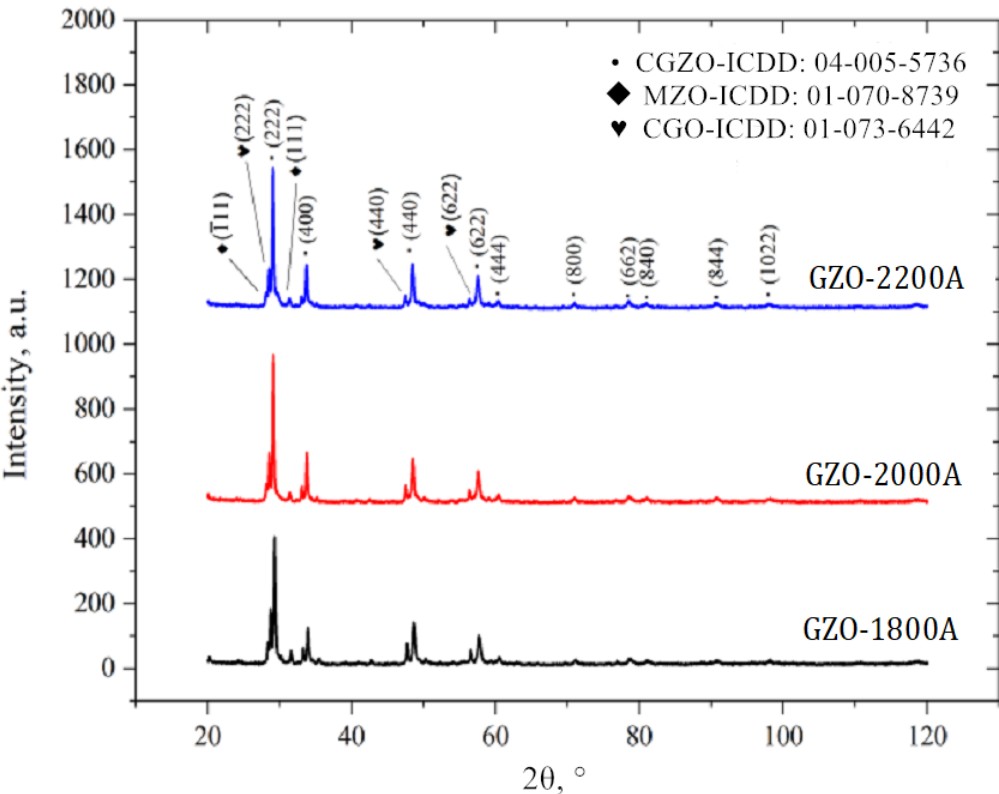

**Figure 3.** XRD patterns of the Gd$_2$Zr$_2$O$_7$ layer. PS-PVD process: produced using different power currents I = 1800, 2000, and 2200 A.

### 3.3. Thermal Properties

Results of the specific heat analyses (Figure 4) and thermal diffusivity (Figure 5) were used to calculate thermal conductivity (Figure 6) while keeping the material density constant. However, in the conductivity calculations, the presence of pores was not considered due to the difficulty of determining their geometry and type. The diffusivity and thermal conductivity results indicate similar values of both of these values for the GZO layers made with power currents of 1800, 2000, and 2200 A. On the other hand, the Gd$_2$Zr$_2$O$_7$ layer produced with the highest power current (GZO-2200A) had a conductivity lower than that of YSZ by an average of 40% in the tested temperature range. The lower thermal conductivity of Gd$_2$Zr$_2$O$_7$ (GZO) in comparison with YSZ was reported by Moskal et al. [38,39].

The obtained values of thermal diffusivity and conductivity (Figures 5 and 6) were similar to results from experiments and models of the quasi-columnar thermal conductivity of GZO and YSZ layers conducted by Qiu et al. [40]. This is grounded in the similarity of the microstructures of ceramic layers made from GZO and YSZ. On the other hand, the content of spheroidal particles from the starting mixture in the microstructure of the obtained GZO layer produced using plasma power currents of 1800, 2000, and 2200 A might have had a significant influence on the differences in thermal conductivity of the gadolinium zirconate layers, but this requires further investigation. There is currently no explanation of the influence of small spheroidal particles on the thermal conductivity of PS-PVD coatings.

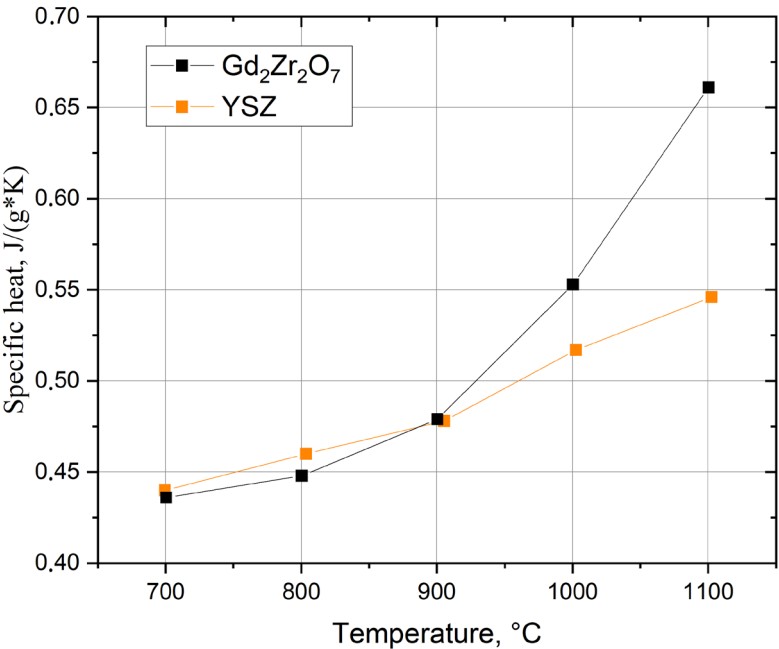

**Figure 4.** Specific heat of $Gd_2Zr_2O_7$ (GZO-2200A) and YSZ (YSZ-2200A) in the temperature range 700–1100 °C.

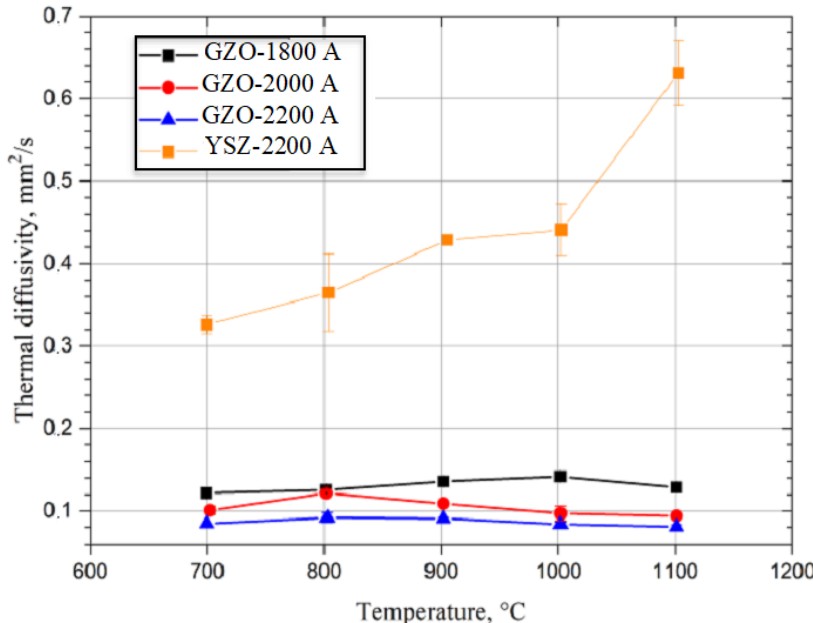

**Figure 5.** Measured diffusivity for YSZ and for $Gd_2Zr_2O_7$ (GZO-1800A, -2000A, and -2200A) layers produced using different power current values.

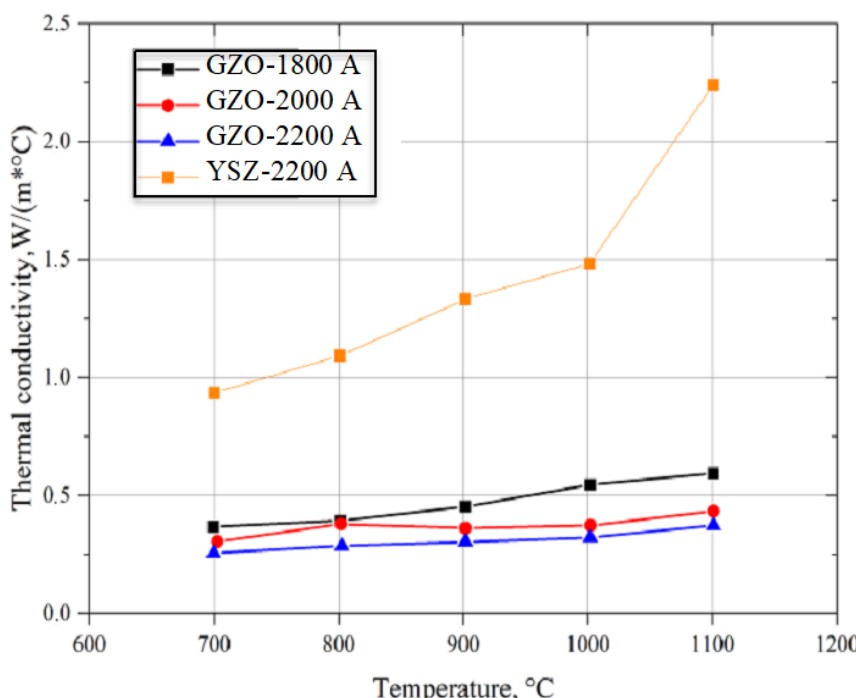

**Figure 6.** Thermal conductivity for YSZ-2200A and for $Gd_2Zr_2O_7$ (GZO-1800A, -2000A, and -2200A) layers produced using different power current values.

## 4. Discussion

The main achievement that we presented in this paper was the synthesis of pyrochlores using the newly developed reactive PS-PVD process, using only a mixture of $ZrO_2$ and $Gd_2O_3$ powders, as well as the production of layers with a columnar structure similar to those produced in the EB-PVD process [35]. The layer made of gadolinium zirconate (GZO) in the 1800, 2000, and 2200 A processes had similar values of thermal diffusivity and thermal conductivity in the temperature range 700–1200 °C, and had thermal diffusivity and thermal conductivity values about 40% lower than those of YSZ [38,39]. This makes it possible to use GZO as a material for the outer ceramic layer of thermal barrier coatings produced in reactive PS-PVD processes with a lower feedstock powder cost. The obtained structure of the coating was similar to the calcinated gadolinium zirconate previously reported by Li [28] and Zhu [29]. The layer produced in the GZO-2200A process is the most promising for further research, as well as for research in terms of its possible application in the aviation and energy industries. The currently developed TBC/EBCs [41] produced in the PS-PVD process might also be achieved using the reactive PS-PVD process.

**Author Contributions:** P.P.—technology and experimental concept and conduction, structure and phase composition analysis; K.D.—measurement of coating thermal properties; M.D.—analysis of thermal properties results; M.G.—conducting of the experimental process, manuscript preparation. All authors have read and agreed to the published version of the manuscript.

**Funding:** This project was co-financed by the European Areal Development Fund under the Operational Programme Innovative Economy and the National Centre for Research and Development Poland (NCBR)— Grant No. INNOLOT/I/7/NCBR/2013—TED.

**Institutional Review Board Statement:** Not applicable.

**Informed Consent Statement:** Not applicable.

**Data Availability Statement:** Not applicable.

**Conflicts of Interest:** The authors declare no conflict of interest.

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
