# Peer review of "Synthesis of Gd2Zr2O7 Coatings Using the Novel Reactive PS-PVD Process"

_coatings, doi:10.3390/coatings11101208_

Round 1
Reviewer 1 Report
This article has successfully deposited Gd2Zr2O7 coatings through novel Reactive PS-PVD method. The result that Gd2Zr2O7 coatings behave 40% lower thermal conductivity than YSZ coatings in 700-1200℃ is very attractive.
However, some questions are needed to be answered:
(1) It is clear that Gd2Zr2O7 coatings have great advantage in thermal conductivity compared with traditional YSZ coatings. However, Fig. 1 shows poor integrity of micro-structures of Gd2Zr2O7 coatings since there are too many large defects and holes. So, it would be a long time before Gd2Zr2O7 coatings came to application. If there be some measures to aculeate the progress?
(2) There is no direct annotation in Fig. 2(c), or is sign (c) just missing?
Author Response
Acording to reviewer recomentation I would like to ask for questions:
(1) It is clear that Gd2Zr2O7 coatings have great advantage in thermal conductivity compared with traditional YSZ coatings. However, Fig. 1 shows poor integrity of micro-structures of Gd2Zr2O7 coatings since there are too many large defects and holes. So, it would be a long time before Gd2Zr2O7 coatings came to application. If there be some measures to aculeate the progress?
The ingegrity is a problem in coatings formation. Hovewer presented article is a brief report of selection of deposition parameters and its influence on structure and themal conductivity. In this case the Gd2Zr2O7 formed using lower power current migh have a problem with corrects structure. During deposition experimental usin 2200A the better structure was obtained. The coatings deposited on samples as well on small turbine blade had no problem with adhesion and spallation of coating was not observed. The poor integrity might be connected with sample preparation procedure - we used typical backelite resin and in might have the influence on coating structure visible on the photos. In further experimental we planning the using a powders with smaller grain size and increasing of power current during spraying
(2) There is no direct annotation in Fig. 2(c), or is sign (c) just missing?
The (c) annotation in Fig 2(c) was missed and was corrected
Reviewer 2 Report
Table 1 doesn’t differentiate YSZ and 2200A as all parameters are the same. It should mention that 2200A is for GZO. It would be good to add labels in Figure 1 corresponding to branch, pore, or fine fraction mentioned in text. Photos of samples and experimental setups should be presented. What are the material densities of YSZ and GZO used to calculate thermal properties? Is the assumption of constant density valid when the porosity may change? Only YSZ in figure 5 shows error bar. Error bar should be added for all measurements in Figures 4-6. There are broken languages throughout the paper, such as line 174. Careful writing is needed. There is no actual validation. The purpose of coating is to prevent high temperature and oxidation. Analysis on microstructure and thermal properties is insufficient to prove the effectiveness of the proposed coating. The coated sample should be put under high temperature with the measurement of temperature and evaluation of oxidation to make a point.Author Response
The article was corrected according to reviewer recomendation
Table 1 doesn’t differentiate YSZ and 2200A as all parameters are the same. It should mention that 2200A is for GZO.
It would be good to add labels in Figure 1 corresponding to branch, pore, or fine fraction mentioned in text.
A: We named the samples as YSZ-2200A and GZO-1800 to GZO-2200 in text and figures
Photos of samples and experimental setups should be presented.
A:In the biref repor so we presenting only selected results. Unfortunetly we do not have the macrophotography of samples
What are the material densities of YSZ and GZO used to calculate thermal properties? Is the assumption of constant density valid when the porosity may change?
The material was calculaded without porosity which was not measured, In can increase a little bit the error of thermal properties measurement
Only YSZ in figure 5 shows error bar. Error bar should be added for all measurements in Figures 4-6.
A: There is not many space for error bar on figures
There are broken languages throughout the paper, such as line 174. Careful writing is needed.
There is no actual validation. The purpose of coating is to prevent high temperature and oxidation. Analysis on microstructure and thermal properties is insufficient to prove the effectiveness of the proposed coating. The coated sample should be put under high temperature with the measurement of temperature and evaluation of oxidation to make a point.
A: The complete cyclic oxidation test at 1100oC was conducted for all samples. In acutal biref report we do not present of all results of our experimental. We planning the new full article of oxidation of other zirconates produced during our experimentals using Reactive PS-PVD process.
Reviewer 3 Report
1) Fig.1 ; explanation of the different figures.
2) How do they connect the unmelted particles with current density?
3) How do they connect the diffusivity with the thermal conductivity?
4) Which factors influence the thermal conductivity?
5) Explain, in details, fig. 5.
Author Response
1) Fig.1 ; explanation of the different figures.
The additional description was added to the fig 1a-d) The presence of spheroidal particles was explained
2) How do they connect the unmelted particles with current density?
3) How do they connect the diffusivity with the thermal conductivity?
According to Q2 and Q3 there in no information about the influence of those spheridical particles on thermal conductivity. Sometimes they removes from the coating or stock between columns. We will analyse it in further articles. Some modification in the article text in this case were made
4) Which factors influence the thermal conductivity?
The main factor of thermal conductivity is thetype of materials and as an secondary - morphology of columns
5) Explain, in details, fig. 5
The small modification of text incl explaination of Fig 5 was added to article
Reviewer 4 Report
The article investigated the microstructure and thermal conductivity of TBC by PSPVD process, and the authors changed the power. I suggest some minor corrections,
- The purpose is obscure in the introduction. The purpose mentioned in section 2 is too general. Microstructure control? conductivity control? or what else? Strengthen your purpose of this article and clarify it.
- Change the "ZrO2 x nY2O3" to ".....mol%(or wt%) Y2O3 doped ZrO2" in the introduction
- Change the symbol of GZO to bold one in the XRD results.
Author Response
The article investigated the microstructure and thermal conductivity of TBC by PSPVD process, and the authors changed the power. I suggest some minor corrections,
- The purpose is obscure in the introduction. The purpose mentioned in section 2 is too general. Microstructure control? conductivity control? or what else? Strengthen your purpose of this article and clarify it.
- A:The article is a brief report and more details of experimental procedure will be described in our next article. We synthetized different pyrochlores and this results will be presented in next full articles. The tiltle of articles is connected with novel Reactive PS-PVD process. In next articles the section 2 will be much more deeper described
- Change the "ZrO2 x nY2O3" to ".....mol%(or wt%) Y2O3 doped ZrO2" in the introduction.
- We changed the ZrO2 x nY2O3 to simple YSZ what is usually for description of this type material. The concentration of dopin elements is different and might by up to about 20 wt %
- Change the symbol of GZO to bold one in the XRD results
- We corrected the all images in article according to two reviewers suggestions
Round 2
Reviewer 2 Report
Comments are addressed properly.